# Dynamic Interactions between Autophagosomes and Lipid Droplets in *Chlamydomonas reinhardtii*

**DOI:** 10.3390/cells8090992

**Published:** 2019-08-28

**Authors:** Quynh-Giao Tran, Hyang Ran Yoon, Kichul Cho, Seon-Jin Lee, José L. Crespo, Rishiram Ramanan, Hee-Sik Kim

**Affiliations:** 1Cell Factory Research Center, Korea Research Institute of Bioscience and Biotechnology (KRIBB), Daejeon 34141, Korea; 2Department of Environmental Biotechnology, KRIBB School of Biotechnology, University of Science and Technology, Daejeon 34113, Korea; 3Immunotherapy Convergence Research Center, KRIBB, Daejeon 34141, Korea; 4Environmental Safety Group, Korea Institute of Science and Technology (KIST) Europe, Campus E 7.1, 66123 Saarbrücken, Germany; 5Environmental Disease Research Center, KRIBB, Daejeon 34141, Korea; 6Instituto de Bioquímica Vegetal y Fotosíntesis, CSIC-Universidad de Sevilla, 41092 Sevilla, Spain; 7Sustainable Resources Laboratory, Department of Environmental Science, Central University of Kerala, Kasaragod 671316, Kerala, India

**Keywords:** autophagy marker, *Chlamydomonas*, chloroquine, lipid droplet, mCherry-ATG8, microalgal lipophagy

## Abstract

Autophagy is a highly conserved catabolic process in eukaryotic cells by which waste cellular components are recycled to maintain growth in both favorable and stress conditions. Autophagy has been linked to lipid metabolism in microalgae; however, the mechanism underlying this interaction remains unclear. In this study, transgenic *Chlamydomonas reinhardtii* cells that stably express the red fluorescent protein (mCherry) tagged-ATG8 as an autophagy marker were established. By using this tool, we were able to follow the autophagy process in live microalgal cells under various conditions. Live-cell and transmission electron microscopy (TEM) imaging revealed physical contacts between lipid droplets and autophagic structures during the early stage of nitrogen starvation, while fusion of these two organelles was observed in prolonged nutritional deficiency, suggesting that an autophagy-related pathway might be involved in lipid droplet turnover in this alga. Our results thus shed light on the interplay between autophagy and lipid metabolism in *C. reinhardtii*, and this autophagy marker would be a valuable asset for further investigations on autophagic processes in microalgae.

## 1. Introduction

Microalgae are plant-like organisms that inhabit freshwater, inter-tidal, marine, and extreme environments. Microalgae are known to accumulate valuable products such as lipids, starch, and carotenoids in response to unfavorable conditions [1]. Thus, treatment of algae with nutrient deprivation, high salinity, extreme temperatures, or high irradiance have become essential strategies for enhancing commercial production of microalgal metabolites [1,2,3,4]. It has been shown that upon exposure to external stressors, eukaryotic cells rapidly trigger a series of systemic responses, which are controlled by autophagy among other factors, to promote survival [5,6]. More specifically, autophagy helps with cleaning and recycling cellular “waste” including protein aggregates and damaged organelles caused by a range of abiotic and biotic stresses in many organisms [5,7,8,9]. Autophagy has been intensively studied in animals and yeasts over the last few decades; however, challenges to expand this research area in microalgae still remain, mostly due to the lack of monitoring tools [9].

In recent years, there has been significant progress in exploring the role of autophagy in the unicellular model *C. reinhardtii* [10,11,12,13,14,15,16,17,18]. For instance, two studies demonstrated that inhibition of autophagic process by treatment with autophagy inhibitors including concanamycin A, bafilomycin A1, and wortmannin reduced the number of lipid droplets accumulated in *C. reinhardtii* cells under nutrient starvation [15,16]. These results suggested that autophagy might be involved in the biogenesis of lipid droplets in this alga. On the contrary, the role of autophagy in lipid degradation was demonstrated in the green microalga *Auxenochlorella protothecoides* when the cells were transferred from heterotrophic to autotrophic growth conditions [19]. How autophagy regulates stress responses in microalgae and how it interacts with algal lipid metabolism in stress conditions remain open questions. A better understanding of this interaction could provide insights to advance the production of biofuel precursors and other valuable metabolites in microalgae.

Autophagic activity can be assessed by observing autophagy-related structures and analyzing the abundance/modification of autophagy-related proteins [20]. Among these proteins, the autophagy-related protein 8 (ATG8) plays a critical role in the formation and maturation of autophagosome in eukaryotic organisms [21]. In *C. reinhardtii*, ATG8 contains an extended 14-amino-acid sequence after the conserved glycine residue (Gly120) at its C-terminus [9]. The nascent ATG8 protein is cleaved at its Gly120 residue by the ATG4 protease to form a cytosolic unconjugated ATG8 protein. Upon autophagy induction, the unconjugated ATG8 becomes conjugated/lipidated to the phospholipid phosphatidylethanolamine (PE) to form ATG8-PE by the action of E1- and E2-like enzymes [9]. Importantly, the ATG8-PE remains associated with autophagosome from the early formation to completed fusion with the lysosome/lytic vacuole, making it a specific marker for monitoring autophagy in vivo [22]. A specific ATG8 antibody has been developed in *C. reinhardtii*, and the lipidation status of this protein can be easily assessed by Western blot analysis [11]. Furthermore, cellular distribution of autophagosomes can be visualized by TEM imaging and immunolocalization assay using ATG8 antibody, although these approaches may require specialized skills and equipment [9]. These methods were the only tools used for setting up the basic knowledge of autophagy-related mechanisms in this alga to date. However, monitoring of autophagic flux and co-localization studies in living algal cells remained entirely unexplored [9].

In this study, we developed *C. reinhardtii* transgenic lines expressing the red fluorescent protein *(mCherry)-ATG8* and investigated the formation of autophagosomes in live algal cells under different conditions. The effect of chloroquine (CQ), an inexpensive lysosomotropic agent, on lytic vacuolar activity and autophagic flux was also examined. In addition, Western blot and TEM analyses were carried out in order to validate autophagic activity in the mutants. By using live-cell imaging, we observed physical interactions between mCherry-labeled autophagosomes and lipid droplets in this green alga under nitrogen starvation. To our knowledge, this provides the first visual evidence for lipid droplet–autophagosome interaction in microalgae.

## 2. Materials and Methods

### 2.1. Microalgal Cultivation

*C. reinhardtii* wild-type strain CC-124 [137c] was grown in Tris-acetate phosphate (TAP) medium [23], in 500 mL conical flasks under continuous illumination of 50 ± 10 µmol m^−2^ s^−1^ at 25 °C, with constant shaking at 90 rpm. When required, a solid medium was prepared by adding 15 g bacto agar per 1 L TAP medium. For nitrogen starvation, cells in exponential phase (approximate cell density 1 × 10^6^ cells mL^−1^) were harvested by centrifugation (2000× *g* for 5 min). Cell pellet was washed once in nitrogen-free medium (TAP-N) before resuspension in TAP-N at the same cell density. For selection of transformants, paromomycin (Sigma-Aldrich, St. Louis, MO, USA) was added to liquid or agar solidified TAP medium at concentration of 25 µg mL^−1^.

### 2.2. Vector Construction

To generate *mCherry-ATG8* fusion construct, the codon-optimized sequence of *mCherry* gene (removed the stop codon) was PCR amplified from the pBR9 mCherry Cr plasmid [24] and cloned into the pET-28a(+) cloning vector as a XhoI/HindIII fragment in front of the *CrATG8* gene. The *RBCS2 intron1* obtained from the pChlamiRNA3int plasmid (Chlamydomonas Resource Center, St. Paul, MN, USA) was cloned as a NdeI/XhoI fragment in front of the *mCherry-ATG8* sequence. Then, the full set (*RBCS2 intron1-mCherry-ATG8*) was cloned back into pChlamiRNA3int to create the pChl-mCherry-ATG8 expression vector (Figure 1).

### 2.3. Generation of mCherry-ATG8 Transgenic Lines

Wild-type cells were transformed by electroporation with GeneArt^®^ MAX Efficiency^®^ Transformation Reagent for Algae protocol and reagent (Invitrogen, Carlsbad, CA, USA). In brief, cells were grown to 1 × 10^6^ cells mL^−1^ in TAP medium as described. Cells were harvested by centrifugation at 2000× *g* for 5 min and washed twice with transformation reagent. Cell pellet was resuspended in transformation reagent to a final concentration of 2 × 10^8^ cells mL^−1^. A total of 1 µg of linearized plasmid was incubated with 250 µL of cells for 5 min on ice. The cell–plasmid mix was then transferred into an ice-cold 0.4 cm–gap cuvette (Bio-Rad, Hercules, CA, USA). Electroporation was performed using the Gene Pulser Xcell^TM^ Total System (Bio-Rad, Hercules, CA, USA) with the following conditions: Resistance of 800 Ω, capacity of 50 µF, field strength of 1.25 kV cm^−1^, and pulse duration of 30 ± 2 ms. Cells were recovered for 14–16 h in 10 mL of TAP supplemented with 40 mM sucrose under dim light at 25 °C, with gentle shaking at 50 rpm and then plated on TAP agar plates supplemented with paromomycin. After 7 days of incubation, single green colonies were transferred to liquid medium containing 25 µg mL^−1^ paromomycin. Stable transformants were obtained after several rounds of selection and subjected to further analyses.

### 2.4. Protein Isolation and Western Blot Analysis

Total cell lysates were prepared with RIPA buffer (100 mM Tris-HCl pH 8.0, 5 mM EDTA, 50 mM NaCl, 50 mM β-glycerophosphate, 50 mM NaF, 0.1 mM Na_3_VO_4_, 1 mM PMSF, 0.5% NP-40, 0.5% sodium deoxycholate, supplemented with 10 µL mL^−1^ Protease Inhibitor Cocktail (Sigma-Aldrich, St. Louis, MO, USA) before use. Same amounts of total protein were denatured by incubation with 5× SDS sample buffer for 5 min in boiling water bath. A total of 30 µg of protein per sample was separated on 15% SDS-PAGE gels at 110 volts and transferred to PVDF membrane (Bio-Rad, Hercules, CA, USA) at 350 mA for 1 h. The anti-CrATG8 polyclonal antibody was kindly provided by Dr. José L. Crespo (Universidad de Sevilla). For Western blot analysis, anti-CrATG8 antibody was diluted 1:2000 in Tris-buffered saline containing 0.1% Tween 20 (TBST buffer) and 5% (*w/v*) skim milk. Anti-rabbit (Abcam, Cambridge, UK) secondary antibodies were diluted 1:10,000 in the same buffer and used to detect ATG8 protein. The Clarity^TM^ Western ECL Substrate (Bio-Rad, Hercules, CA, USA) was used to generate signals and images were captured by Bio-Rad ChemiDoc^TM^ MP System.

### 2.5. Expression Analysis by Real-Time RT-PCR

Cell culture was harvested and cell pellet was snap frozen with liquid nitrogen. Total RNA was extracted using TRIzol^®^ Reagent (Ambion, Austin, TX, USA) and purified with QIAGEN RNeasy Mini Kit in accordance with the manufacturer’s protocol. Genomic DNA contamination was eliminated from total RNA sample using RQ1 RNase-Free DNase kit (Promega, Madison, WI, USA), and RNA purity and quantity were checked with the NanoPhotometer^®^ P360 device (Implen, Munich, Germany). A total of 500 ng of total RNA was converted to first-strand cDNA with oligo(dT) primer using GoScript^TM^ Reverse Transcription System (Promega, Madison, WI, USA) in a 20 µL reaction. The resulting cDNA was used as template for real-time RT-PCR using iQ™ SYBR^®^ Green Supermix (Bio-Rad, Hercules, CA, USA) with the following primers: mCherry_F (5′-ACATCAAGCTGGACATCACC-3′) and mCherry_R (5′-CTTGTACAGCTCGTCCATGC-3′). A total of 10 µL of each RT-PCR product was resolved by electrophoresis on a 1% (*w/v*) agarose gel, and the gel image was captured with a Kodak Gel Logic 100 Digital Imaging System (Kodak, Rochester, NY, USA).

### 2.6. Confocal Microscopic Analysis

For lytic vacuole staining, exponential phase cells treated with autophagic stimuli (rapamycin or nitrogen depletion) were harvested by centrifugation at 2000× *g* for 5 min and placed in 1 mL of the same fresh medium. LysoSensor Green DND-189 (stock 1 mM in DMSO) (Life Technologies, Carlsbad, CA, USA) was then added to the sample at the final concentration of 5 µM and incubated for 30 min in the dark at 37 °C according to the manufacturer’s protocol. At the end of the incubation, cells were directly observed under the confocal facility without further treatment. For lipid droplet staining, cells were harvested and stained with 1 µg mL^−1^ BODIPY 505/515 (Invitrogen, Carlsbad, CA, USA) as described previously [17].

Images were obtained by using a Zeiss LSM510 meta-laser scanning confocal microscope (Carl Zeiss AG, Oberkochen, Germany) fitted with 100× objectives and equipped with Nikon camera. Excitation/emission maxima of 587/610 nm for mCherry, 655/667 nm for chlorophyll *a*, 448/505 nm for LysoSensor Green DND-189, and 505/515 nm for BODIPY 505/515 were used to acquire fluorescence images. Images were processed using Zeiss LSM510 software (Carl Zeiss AG, Oberkochen, Germany) and quantified using ImageJ software (NIH, Bethesda, MD, USA) when necessary.

### 2.7. Transmission Electron Microscopy (TEM) Analysis

The cells were fixed in 2.5% paraformaldehyde–glutaraldehyde mixture buffered with 0.1 M phosphate (pH 7.2) for 2 h, post fixed in 1% osmium tetroxide in the same buffer for 1 h, dehydrated in graded ethanol and propylene oxide, and embedded in Epon-812. Ultra-thin sections, made by ULTRACUT E (Leica, Wetzlar, Germany) ultramicrotome, were stained with uranyl acetate and lead citrate and examined under CM 20 (Philips, Amsterdam, The Netherlands) electron microscope.

### 2.8. Flow Cytometry Analysis

Flow cytometry analysis was performed using the BD FACS Aria^TM^ Cell Sorters (BD Biosciences, San Jose, CA, USA) equipped with 488 nm (blue) and 633 nm (red) lasers. Data was analyzed with FlowJo^TM^ software version 7.6.5 (FlowJo LLC, Ashland, OR, USA). A minimum of 50,000 algal cells per sample were recorded for fluorescence analysis. Experiments were repeated three times and mean values of fluorescence intensity are shown.

### 2.9. Quantification of Autophagic Structures

In order to quantify the number and size of mCherry-labeled autophagic structures, confocal images were processed and analyzed using ImageJ software (version 1.46, NIH, Bethesda, MD, USA). The number and size of autophagic structures per cell were automatically measured using the Analyze Particles function in ImageJ after setting the threshold (only particles larger than 5 pixels were included). In Figure 3, the lipid droplets positive with mCherry-ATG8 were defined as structures which labeled with both red and green fluorescence. Co-localization analysis was done using the co-localization plugin in ImageJ (http://imagej.net/mbf/colour_analysis.htm) and the number of co-localized structures were counted per cell. For all experiments, approximately 50–80 cells per condition were analyzed in at least five independent confocal images per condition.

### 2.10. Statistical Analyses

All experiments were performed in triplicate and data are presented as the means ± standard deviation (SD). Statistical analyses were done using Origin Pro 9.0 software (OriginLab, Northampton, MA, USA). One-way ANOVA and subsequent Tukey’s post-hoc *t* test or Mann-Whitney U test was used to analyze the statistical significance (**, *p* < 0.01) of the data.

## 3. Results

### 3.1. Autophagic Responses in C. reinhardtii Transgenic Lines Expressing mCherry-ATG8

To monitor autophagy flux in *C. reinhardtii*, stable transgenic lines expressing a red fluorescent protein (mCherry)-tagged ATG8 fusion construct were generated. For this, the nuclear codon-optimized *mCherry* gene was fused to the full-length *CrATG8* gene at the N terminus [25]. The *mCherry-ATG8* fusion construct was expressed in *C. reinhardtii* using pChlamiRNA3int (pChl) expression system. This vector was designed for miRNA-mediated gene silencing in *C. reinhardtii* [26]. Herein, the artificial miRNA precursor was removed, and *mCherry-ATG8* was cloned between *PSAD* promoter and *PSAD* terminator (see Materials and Methods section, Figure 1A). All putative transformants carrying the pChl-mCherry-ATG8 construct (hereafter referred to as mCherry-ATG8 transgenic lines) were confirmed by genomic PCR after several rounds of selection on paromomycin (*APHVIII* gene), and the expression of *mCherry-ATG8* was analyzed using Real-time RT-PCR (Figure 1B). In addition, mCherry fluorescence intensity of transgenic lines were analyzed by flow cytometry (Figure 1C). Two transgenic lines (#5 and #8) showed the same growth characteristics from wild-type while exhibiting detectable levels of mCherry fluorescence (Figure 1B–D). Therefore, they were selected for further biochemical analyses, the pictures of transgenic line #8 were presented in the study.

Confocal fluorescence microscope analyses revealed that mCherry-ATG8 diffused throughout the cytoplasm in cells growing under optimal conditions but appeared as bright dots after autophagy was induced by rapamycin treatment. No background fluorescence was detected in wild-type cells at the excitation/emission wavelengths of mCherry, indicating the specificity of the mCherry signal (Figure 1E). Although further confirmation would be necessary, these vesicles would likely correspond to autophagosomal structures, as rapamycin has been proven to induce the formation of autophagosomes in various species [27,28].

It is a concern that both native ATG8 and the fluorescence-ATG8 fusion protein resulted from ectopic expression tend to be aggregated into intracellular inclusion bodies [29]. Thus, punctate structures positive with ATG8 or fluorescent-tagged ATG8 may correspond to either aggregates or functional autophagosomes [29]. To verify the nature of mCherry-ATG8 structures in this study, transgenic cells were examined by fluorescent confocal microscopy after 12 h of nitrogen starvation in the presence or absence of CQ (Figure 2). It has been shown that CQ inhibits autophagy by neutralizing the lysosomal/vacuolar pH, thereby blocking the activity of hydrolytic enzymes and the subsequent steps in autophagy [22]. While concentrations up to 100 µM of CQ did not affect cell viability, the ATG8 protein was more abundant and the lipidated form ATG8-PE was detectable in cells treated with 100 µM CQ, indicating that a dose of 100 µM CQ was sufficient to inhibit autophagic flux in *C. reinhardtii* (Figure 2A).

To investigate the role of lytic vacuoles in autophagic degradation in this alga, LysoSensor Green DND-189 (LSG) dye, a fluorescent probe that targets lysosomes/lytic vacuoles in various cell types, was used [30]. Flow cytometry analysis revealed a marked increase in LSG fluorescent intensity from 21.4 ± 1.7 arbitrary units (A.U.) of control (Rapamycin-/CQ-) to 36.5 ± 7.3 A.U. of rapamycin-treated cells (Rapamycin+/CQ-), corresponding to an increased formation of acidic lytic vacuoles (Figure 2B). This result is consistent with a previous research showing that rapamycin treatment triggered autophagy, leading to the accumulation of lytic vacuoles in *C. reinhardtii* [10]. In contrast to many vacuolar marker probes, LSG is pH sensitive and only exhibits fluorescence in acidic compartments [30]. As expected, treatment with CQ resulted in a reduced LSG fluorescent intensity (20.9% and 33.1% of control and rapamycin-treated cells, respectively) (Figure 2B). These results indicated that LSG and CQ can be used to assess the vacuolar status in *C. reinhardtii*, and treatment with CQ caused a shift in vacuolar pH in this alga. In addition, Western blot analysis revealed a significant increase in ATG8-PE level in cells co-treated with rapamycin and CQ (Figure 2B), which confirmed that CQ can effectively inhibit autophagic degradation in *C. reinhardtii* [15].

To determine whether mCherry-ATG8 labeled autophagosomes in a specific manner, transgenic cells were subjected to nitrogen starvation to trigger autophagy and the distribution of mCherry-labeled structures was observed (Figure 2C). Upon nitrogen starvation, concurrent LSG staining revealed several enlarged yellow dots, which labeled with both mCherry and LSG, indicating that these structures are likely ‘microalgal autolysosomes’, a fusion compartment between autophagosome and lytic vacuole (Figure 2C, upper panels). The mCherry-labeled structures in this condition were relatively larger but fewer in number compared to those in the presence of CQ (Figure 2C–E). On the contrary, mCherry-labeled structures accumulated more numerously and had a less distinct size when vacuolar degradation was inhibited by CQ. These observations revealed dynamic changes in cellular distribution of mCherry-ATG8, thus, indicating that this fusion protein may constitute a good marker for the visualization of autophagic structures in *C. reinhardtii*. Overall, the present study provides evidences that autophagy is integrated with stress responses and the lytic vacuole could serve as the last destination for autophagic degradation in this microalga.

### 3.2. Live Cell Imaging Revealed Interactions between Autophagosomes and Lipid Droplets in C. reinhardtii

Algae usually tend to accumulate neutral lipids in a distinctive organelle called lipid droplet upon exposure to various stress conditions, especially nitrogen starvation. Under such conditions, autophagy is also induced, allowing the recycling of cellular components to support cell survival [15,31,32]. Although autophagy has been linked to the degradation of lipid droplets through a process termed lipophagy in mammals and yeast, the role of autophagy in lipid metabolism in microalgae remained to be explored. Herein, we investigated the potential interplay between autophagosomes and lipid droplets in *C. reinhardtii* using the mCherry-ATG8 marker. Live imaging revealed dynamic interactions between mCherry-labeled structures and lipid droplets (stained with BODIPY 505/515) throughout nitrogen starvation (Figure 3, Appendix A). In the early stage of stress (first 12 h), the newly formed autophagosomes were often found alongside the lipid droplets (Figure 3A, Appendix A). TEM analysis was used in an attempt to clarify the autophagosomes-lipid droplets interaction in the early stage of starvation [33]. As shown in Figure 3B, double-membrane vesicles that may correspond to autophagosomes were seen next to a lipid droplet in cells starved for 12 h. It is interesting to note that we could not observe any fusion events between autophagosomes and lipid droplets in starved cells at 12 h post-starvation. The interaction between these two organelles at this stage remained to be explored, although it has been reported that normal autophagic flux is required for the formation of lipid droplets in *C. reinhardtii* [15,16].

In addition to the above findings, we detected a few punctate structures labeled with both mCherry-ATG8 and BOPIDY after prolonged periods of starvation (24 h) (Appendix A). The number of yellow (red + green) punctates was significantly increased in starved cells at 48 h post-starvation (Figure 3C). Confocal microscopic image of a *C. reinhardtii* cell expressing mCherry-ATG8 starved for 48 h (Figure 3D). The fluorescence intensity profiles taken along the dashed lines indicated distinct phases of autophagosome–lipid droplet fusion (Figure 3E,F). For example, a lipid droplet (green) can be easily distinguished with an autophagosome–lipid droplet fusion (yellow) or an autophagosome (red) based on their intensity profiles. A three-dimensional (3D) rendering from Z-stacks of the cell in Figure 3D is available in Appendix A. Taken together, these data suggested that autophagosomes might be in association with microalgal lipid droplets through a process similar to lipophagy, indicating by the fusion between autophagosomes and lipid droplets at the later stages of nitrogen starvation. It remains to be explored how physical interaction occurs between autophagosomes and lipid droplets, and the relative contribution of autophagy-related pathway versus lipase-mediated degradation of lipid droplets in microalgae.

## 4. Discussion

In recent years, studies of autophagy in photosynthetic organisms including land plants and microalgae have been greatly expanded. Recent findings suggested that autophagy might play a pivotal role in triacylglycerols biosynthesis and lipid droplets formation in *C. reinhardtii* [15,16]. However, there is no confirmatory evidence on the same as well as the role of lipophagy in lipid turnover in microalgae [19]. By using the mCherry-ATG8 marker, we were able to visualize the autophagy process in living *C. reinhardtii* cells grown under various conditions.

It is now known that plants may have distinct types of selective autophagy which might not be conserved in non-photosynthetic organisms [34]. Since *C. reinhardtii* is a model organism that retains important features of both plants and animals, a better understanding of autophagy in this alga would constitute the missing link between these two eukaryotic kingdoms [35,36]. Previous works have tentatively suggested that endogenous ATG8 diffusely distributes in the cytoplasm of *C. reinhardtii* cells under ideal conditions but relocates to punctate structures that might represent autophagosomes upon the induction of autophagy [9]. Similar localization patterns were observed in the present study in *C. reinhardtii* using the mCherry-ATG8 marker. The degradation of autophagosome and its cargo has been known to occur inside the autolysosome, a fused structure between an autophagosome and a lysosome containing hydrolytic enzymes, in animal cells [37]. By using TEM imaging, a previous study has shown that autophagic bodies are accumulated within the vacuoles in *C. reinhardtii* cells with impaired lytic function [15]. In this study, we indeed observed significant colocalization of lytic vacuoles with mCherry-labeled autophagosomes under nitrogen starvation, confirming that the vacuoles are involved in autophagic degradation in this alga (Figure 2).

Over the last two decades, lipid droplets have received intensive focus [38]. Lipid droplets not only serve as storage depots for cellular lipids, but also participate in multiple physiological pathways, including protein storage, membrane transport and the replication of pathogenic viruses [39]. The breakdown of neutral lipids core in lipid droplets is primarily accomplished by lipolysis, which involves a variety of cytosolic lipases [39]. In addition to the lipolysis pathway, recent studies have indicated the participation of autophagy in lipid droplet turnover through a process known as lipophagy in mouse, rice, yeast, and microalgae [19,40,41,42,43,44]. By using mCherry-ATG8 tool, we confirm the physical interactions between autophagosomes and lipid droplets in *C. reinhardtii* (Figure 3). At the early stage of starvation, autophagy may supply precursors for lipid droplets biogenesis in microalgae through an unexplored mechanism, as we observed that autophagosomes maintained close contact but did not fuse to lipid droplets. Indeed, nutrient recycling via autophagy has been reported to play a crucial role in the regeneration of precursors for the formation of lipid droplets in *C. reinhardtii* [15,16]. Recently, autophagy has been shown to play a dual role in controlling lipid synthesis and degradation in the model plant *Arabidopsis thaliana* [45]. It would be interesting to explore the interplay between autophagosomes and lipid droplets in microalgae. The fusion of autophagosomes and lipid droplets at the later stages of starvation suggests the involvement of an autophagic pathway in the breakdown of lipid droplets in *C. reinhardtii* to sustain cellular homeostasis and promote cell survival. It remains to be elucidated how autophagosomes sequester lipid droplets in this alga. Taken together, it is likely that under stress conditions, the formation and degradation of both autophagosomes and lipid droplets occur in a hierarchical and balanced fashion in microalgae. Thus, understanding the molecular mechanism between autophagy and lipid metabolism will open up new avenues of biofuels production from these unicellular organisms.

## Figures and Tables

**Figure 1 cells-08-00992-f001:**
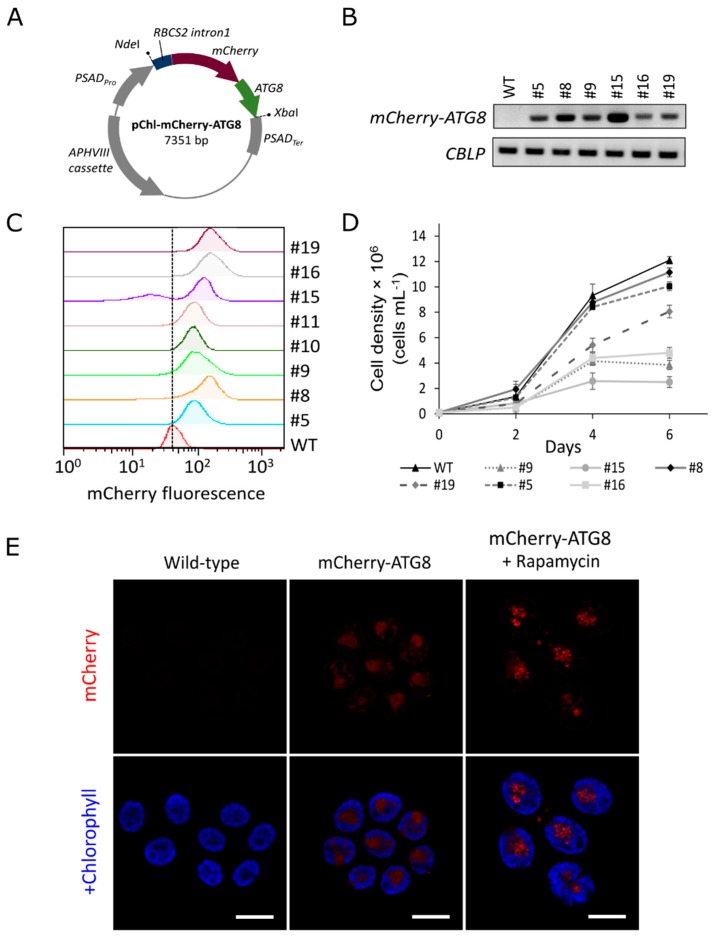
Generation of *C. reinhardtii* transgenic lines expressing the red fluorescent protein (mCherry)-ATG8. (**A**) Schematic drawing of the pChl-mCherry-ATG8 vector for microalgal transformation. (**B**) Real-time RT-PCR analysis. A total of 10 μL of PCR products were separated by electrophoresis and gel image are shown. (**C**) Flow cytometry analysis of transgenic lines. A vertical dashed line is provided for visual reference. (**D**) Comparison of growth rates. Numbers indicated independent transgenic lines; WT, wild-type. (**E**) Confocal microscopic imaging of *C. reinhardtii* cells expressing mCherry-ATG8. Under normal growth condition, mCherry-ATG8 (red) diffused throughout the cytoplasm in transgenic cells. Upon autophagy induction by rapamycin (500 nM) treatment for 16 h, mCherry-ATG8 labeled vesicles appeared as bright dots. No mCherry fluorescence was detected in wild-type cells, indicating the specificity of mCherry signal. Chlorophyll *a* fluorescence (blue) serves as reference for cell size and morphology. Results are representative images of three replicates. Bars, 10 μm. *PSAD_Pro_*, *PSAD* promoter; *Nde*I or *Xba*I, restriction sites; *RBCS2 intron1*, first intron of *RBCS2* gene in *C. reinhardtii*; *APHVIII*, paromomycin-resistance gene; *PSAD_Ter_*, *PSAD* terminator.

**Figure 2 cells-08-00992-f002:**
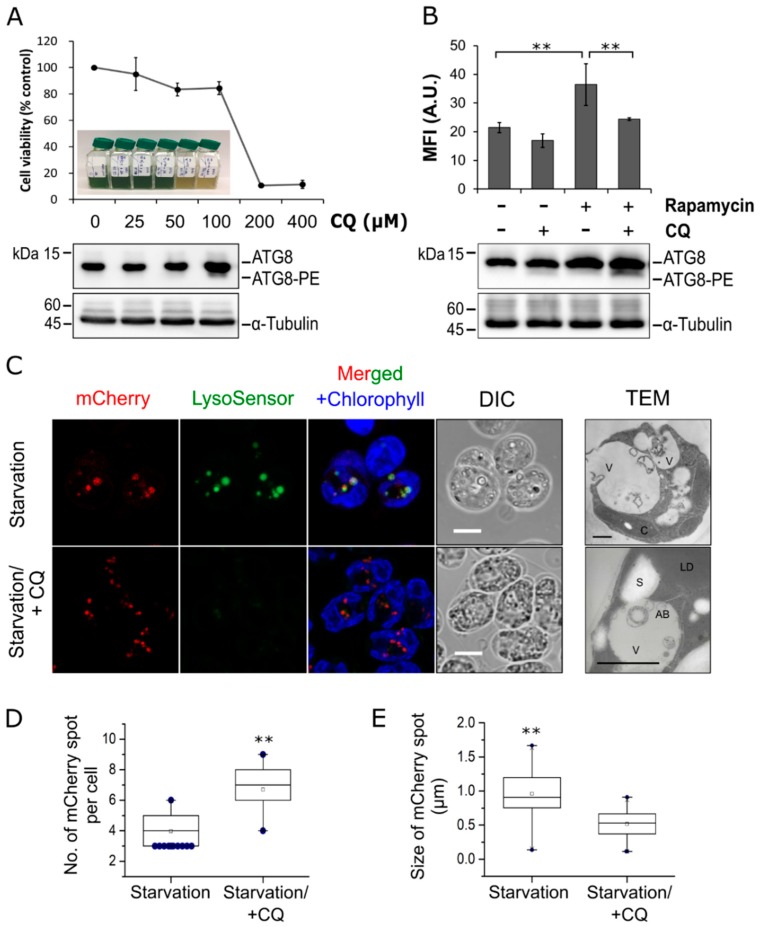
Influence of chloroquine on vacuolar acidification and autophagy activity in *C. reinhardtii*. (**A**) Cell viability and the corresponding ATG8 protein levels in *C. reinhardtii* cells treated with different concentrations of chloroquine (CQ) for 48 h. The Western blot data for cells treated with 200 μM and 400 μM CQ are not available due to extensive cell death. The unconjugated form ATG8 and conjugated/lipidated form ATG8-PE are indicated. α-Tubulin are shown as a loading control. (**B**) Mean fluorescence intensity (MFI) of LysoSensor Green DND-189 (LSG) and Western blot analysis of ATG8 in cells treated with rapamycin in the absence (−) or presence (+) of 100 µM CQ for 16 h. A total of 100,000 cells were acquired for flow cytometry analysis per replicate per condition. (**C**) Co-localization of autophagosomes and lytic vacuoles. Cells expressing mCherry-ATG8 were subjected to nitrogen starvation for 12 h in the absence or presence of 100 µM CQ, followed by staining with LSG. mCherry-ATG8 (red), LSG (green), and merged channels of mCherry-ATG8, LSG, and chlorophyll *a* (blue) are shown. (**D**,**E**) Quantification of the number (**D**) or size (**E**) of structures labeled with mCherry in cells treated as in (**C**). Box plots indicate the medians, means, and quartiles while filled dots represent the outliers of each data set. Statistical significance was analyzed using one-way ANOVA and subsequent Tukey’s post-hoc *t* test and *P*-values obtained are indicated (**, *p* < 0.01). Bars, 5 μm.

**Figure 3 cells-08-00992-f003:**
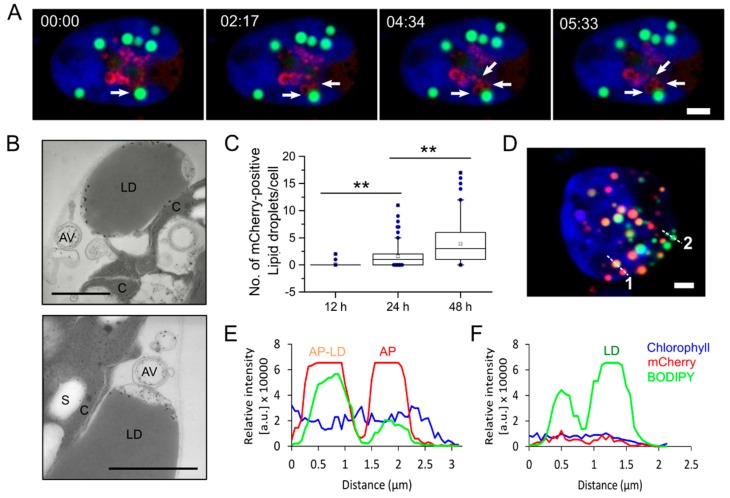
Dynamic interactions between autophagosome and lipid droplets in *C. reinhardtii*. (**A**) Still frames taken from Appendix A showed that mCherry-ATG8 labeled structures were in close proximity to a lipid droplet (LD) in algal cells starved for nitrogen for 12 h. (**B**) TEM images of cells grown under same condition as in (**A**) showing autophagic vacuoles near LDs. (**C**) Quantification of LD positive with mCherry-ATG8 at 12 h, 24 h, and 48 h of nitrogen starvation. The number of vesicles labeled with both red and green fluorescence was counted per cell (50–80 cells per condition) using the Co-localization plugin (ImageJ). Box plots indicate the medians, means and quartiles. Outliers of each data set are presented as filled dots. Statistical significance was analyzed using the Mann-Whitney U test and P-values obtained are indicated (**, *p* < 0.01). (**D**) Representative image of a cell starved for 48 h. mCherry-ATG8 (red), BODIPY 505/515 (green), and merged channels of mCherry-ATG8, BODIPY, and chlorophyll *a* (blue) are shown. (**E**,**F**) Fluorescence intensity profiles along the dashed line 1–2 in (**D**), respectively. Bars, 2 μm in (**A**,**D**) or 1 µm in (**B**). AV, autophagic vacuole; LD, lipid droplet; C, chloroplast; S, starch granule; AP–LD, fusion between an autophagosome and a lipid droplet; AP, autophagosome.

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
