# Peer review of "Dynamic Interactions between Autophagosomes and Lipid Droplets in Chlamydomonas reinhardtii"

_cells, 2019, doi:10.3390/cells8090992_

Round 1
Reviewer 1 Report
This manuscript describes the establishment of a novel molecular tool to study autophagous flux in Chlamydomonas reihhardtii.
Previous studies have contributed to understanding the role of autophagy in this organism using microscopic, biochemical and histochemical techniques and inhibitors.
The authors successfully expressed the N-terminally fused mCherry-ATG8 and detected labeled punctate structures corresponding to autophagosomes and importantly documented the interaction of labeled structures with lipid droplets.
The data were supported with application of rapamycin and an inhibitor of vacuolar acidification chloroquine.
Consistent with the several emerging reports, the dual role of autophagy in lipid droplet biogenesis in microalgae, particularly under N starvation, and degradation following nutrient supplementation was revealed.
I have only a few remarks to this nice manuscript. Some clarification for Fig. 1 is required. In Fig. 1A, figures above the SDS-PAGE gel seem to indicate CQ concentration. This needs to be specified. ATG-PE was detected at 100 μM CQ as can be expected for inhibition of vacuolar acidification, but it is less evident in Fig. 1B. I would recommend citing a recent paper where a dual role of autophagy in lipid droplet metabolism was established. “Dual Role for Autophagy in Lipid Metabolism in Arabidopsis” http://www.plantcell.org/content/31/7/1598
Author Response
Response to Reviewer 1 Comments
Point 1: Some clarification for Fig. 1 is required. In Fig. 1A, figures above the SDS-PAGE gel seem to indicate CQ concentration. This needs to be specified.
Response 1: Thank you for your suggestion. We have considered your comment carefully and thought that the comment is more likely to refer to Fig. 2A, which presents cell viability and ATG8 protein levels in cells treated with CQ. We have added a more detailed description to the figure caption as follow:
L233-236: “Figure 2. Influence of chloroquine on vacuolar acidification and autophagy activity in C. reinhardtii. (A) Cell viability and the corresponding ATG8 protein levels in C. reinhardtii cells treated with different concentrations of chloroquine (CQ) for 48 h. The western blot data for cells treated with 200 μM and 400 μM CQ are not available due to extensive cell death.”
Point 2: ATG-PE was detected at 100 μM CQ as can be expected for inhibition of vacuolar acidification, but it is less evident in Fig. 1B.
Response 2: Thank you for your comment. In Fig. 2A, cells were treated with CQ for 48 h while in Fig. 2B cells were treated with rapamycin in the absence or presence of CQ for 16 h. Thus, the difference in treatment time would contribute to the difference in ATG8-PE levels in two figures. We have added the treatment time for each experiment to the figure caption to make it clearer.
Point 3: I would recommend citing a recent paper where a dual role of autophagy in lipid droplet metabolism was established. “Dual Role for Autophagy in Lipid Metabolism in Arabidopsis” http://www.plantcell.org/content/31/7/1598
Response 3: Thank you for your suggestion. We have read the recommended reference and updated our discussion part as follow:
L355-356: “Recently, autophagy has been shown to play a dual role in controlling lipid synthesis and degradation in the model plant Arabidopsis thaliana [43].”
Reference:
Fan, J.; Yu, L.; Xu, C. Dual Role for Autophagy in Lipid Metabolism in Arabidopsis. Plant Cell 2019, 31, 1598-1613.
Point 4: English language and style are fine/minor spell check required
Response 4: We have carefully read and corrected all the typing mistakes in the entire manuscript. We are very grateful for your detailed comments and suggestions. We hope that the revisions made to our manuscript are sufficient enough to warrant its publication to the journal.

Reviewer 2 Report
20190820 – Review – Tran et al - Cells Dynamic interactions between autophagosomes and lipid droplets in Chlamydomonas reinhardtii
- This is a well crafted and succinct study of the in vivo localization of microbodies containing the ATG8 reporter fused to an mCherry fluorescent protein.
- The authors show in vivo localization of autophagosomal microbodies containing this ATG8 reporter and contrast it to bodily stained lipid droplets in C. reinhardtii cells.
- The key implication of the findings presented here is that these autophagosomes and lipid droplets interact and co-localize to some extent during prolonged starvation, suggesting that lyric processes likely play roles in bio-molecule turnover in these lipid droplets.
- I recommend this paper be accepted for publication with only minor revisions.
- First, there are other types of microbodies in C. reinhardtii, notably the recently characterized peroxisomal microbodies. It would be good if the authors could show one co-localization experiment with their ATG8-mCherry marker and a known peroxisomal targeting peptide fused to another fluorescent reporter.
- Although the authors cite previous characterization of the ATG8 protein as an autophagosmal marker, description of this needs to be included in the introduction so that the reader doesn’t need to read to the end of the manuscript to understand this.
- Minor point: Line 328-329 should read “lipid droplet turnover” not “lipid droplets turnover”
Author Response
Response to Reviewer 2 Comments
Point 1: There are other types of microbodies in C. reinhardtii, notably the recently characterized peroxisomal microbodies. It would be good if the authors could show one co-localization experiment with their ATG8-mCherry marker and a known peroxisomal targeting peptide fused to another fluorescent reporter
Response 1: Thank you for very interesting suggestion. We are currently investigating the interaction between autophagosomes and other cellular organelles including peroxisomes and ER membrane using the ATG8-mCherry marker and other reporters. The data shall be included in our subsequent manuscripts.
Point 2: Although the authors cite previous characterization of the ATG8 protein as an autophagosmal marker, description of this needs to be included in the introduction so that the reader doesn’t need to read to the end of the manuscript to understand this.
Response 2: We agree with the reviewer’s opinion. For a better understanding of the journal’s readership, we have provided additional information in the introduction part as follow:
L59-75: “Autophagic activity can be assessed by observing autophagy-related structures and analyzing the abundance/modification of autophagy-related proteins [20]. Among these proteins, the autophagy-related protein 8 (ATG8) plays a critical role in the formation and maturation of autophagosome in eukaryotic organisms [21]. In C. reinhardtii, ATG8 contains an extended 14-amino-acid sequence after the conserved glycine residue (Gly120) at its C-terminus [9]. The nascent ATG8 protein is cleaved at its Gly120 residue by the ATG4 protease to form a cytosolic unconjugated ATG8 protein. Upon autophagy induction, the unconjugated ATG8 becomes conjugated/lipidated to the phospholipid phosphatidylethanolamine (PE) to form ATG8-PE by the action of E1- and E2-like enzymes [9]. Importantly, the ATG8-PE remains associated with autophagosome from the early formation to completed fusion with the lysosome/lytic vacuole, making it a specific marker for monitoring autophagy in vivo [22]. A specific ATG8 antibody has been developed in C. reinhardtii, and the lipidation status of this protein can be easily assessed by Western blot analysis [11]. Furthermore, cellular distribution of autophagosomes can be visualized by TEM imaging and immunolocalization assay using ATG8 antibody, although these approaches may require specialized skills and equipment [9]. These methods were the only tools used for setting up the basic knowledge of autophagy-related mechanisms in this alga to date. However, monitoring of autophagic flux and co-localization studies in living algal cells remained entirely unexplored [9].”
Reference:
Mizushima, N.; Yoshimori, T.; Levine, B. Methods in mammalian autophagy research. Cell 2010, 140, 313-326. Nakatogawa, H.; Ichimura, Y.; Ohsumi, Y. Atg8, a ubiquitin-like protein required for autophagosome formation, mediates membrane tethering and hemifusion. Cell 2007, 130, 165-178. Klionsky, D.J.; Abdelmohsen, K.; Abe, A.; Abedin, M.J.; Abeliovich, H.; Acevedo Arozena, A.; Adachi, H.; Adams, C.M.; Adams, P.D.; Adeli, K., et al. Guidelines for the use and interpretation of assays for monitoring autophagy (3rd edition). Autophagy 2016, 12, 1-222.
Point 3: Minor point: Line 328-329 should read “lipid droplet turnover” not “lipid droplets turnover”
Response 3: Thank you for your detailed review of our manuscript. We have edited the writing accordingly.
Point 4: English language and style are fine/minor spell check required
Response 4: We have carefully read and corrected all the typing mistakes in the entire manuscript. We are very grateful for your careful reading of our manuscript and constructive comments. We hope that you will find satisfaction with our responses and the revised manuscript.

Reviewer 3 Report
The paper titled "Dynamic interactions between autophagosomes and lipid droplets in Chlamydomonas reinhardtii" by Tran et al reports their observation on the interaction between lipid droplet and autophagosomes. The paper provides novel observations, however it would have been substantially consolidated if at least two transgenics lines have been studied in detail, instead of only one? A minor suggestion: it will be necessary to give a bit more details in the figure legend, and also for the supplemental data. For the moment, it is not written what is the significance of the colour code etc.
Author Response
Response to Reviewer 3 Comments
Point 1: The paper provides novel observations, however it would have been substantially consolidated if at least two transgenics lines have been studied in detail, instead of only one?
Response 1: Thank you for your comment. As we have mentioned in L198-201, two transgenic lines (#5 and #8) were selected for all biochemical and microscopic analyses. It is worthwhile mentioning that both transgenic lines displayed similar growth rate and autophagic behaviors under testing conditions. In order to reduce the complexity of the manuscript and make it more reader-friendly, the pictures of line #8 were presented in the figures.
Point 2: A minor suggestion: it will be necessary to give a bit more details in the figure legend, and also for the supplemental data. For the moment, it is not written what is the significance of the colour code etc.
Response 2: Thank you for your detailed review of our manuscript. We have carefully checked and revised the figure legends. For instance, we added information regarding the color code of microscopic images to each figure legend. We also provided additional explanations on experimental conditions to the captions of Fig.2 and Supplementary Videos.
Point 3: English language and style are fine/minor spell check required
Response 3: Thank you very much for your encouraging and constructive comments on our manuscript. Regarding English language and style, we have carefully corrected all the typing mistakes in the entire manuscript. We hope that our responses and the revised manuscript will meet your expectations.
